# An image-guided microfluidic system for single-cell lineage tracking

**Mahmut Aslan Kamil[1◉], Camille Fourneaux[2◉], Alperen Yilmaz[3], Stavrakis Stavros[1], Romuald Parmentier[4], Andras Paldi [4], Sandrine Gonin-Giraud[2], Andrew J. deMello[1], Olivier Gandrillon [2,5]***

**1** Institute for Chemical and Bioengineering, Department of Chemistry and Applied Biosciences, ETH Zürich, Zürich, Switzerland, **2** Laboratory of Biology and Modelling of the Cell, Université de Lyon, Ecole Normale Supérieure de Lyon, CNRS, UMR5239, Université Claude Bernard, Lyon, France, **3** Faculty of Medicine, Koç University, Istanbul, Turkey, **4** Ecole Pratique des Hautes Etudes, St-Antoine Research Center, Inserm U938, PSL Research University, Paris, France, **5** Inria, France

◉ These authors contributed equally to this work.
* Olivier.gandrillon@ens-lyon.fr

**Data Availability Statement:** The datasets supporting the conclusions of this article are available at https://www.ncbi.nlm.nih.gov/bioproject/PRJNA882740. The R scripts are

## Abstract

Cell lineage tracking is a long-standing and unresolved problem in biology. Microfluidic technologies have the potential to address this problem, by virtue of their ability to manipulate and process single-cells in a rapid, controllable and efficient manner. Indeed, when coupled with traditional imaging approaches, microfluidic systems allow the experimentalist to follow single-cell divisions over time. Herein, we present a valve-based microfluidic system able to probe the decision-making processes of single-cells, by tracking their lineage over multiple generations. The system operates by trapping single-cells within growth chambers, allowing the trapped cells to grow and divide, isolating sister cells after a user-defined number of divisions and finally extracting them for downstream transcriptome analysis. The platform incorporates multiple cell manipulation operations, image processing-based automation for cell loading and growth monitoring, reagent addition and device washing. To demonstrate the efficacy of the microfluidic workflow, 6C2 (chicken erythroleukemia) and T2EC (primary chicken erythrocytic progenitors) cells are tracked inside the microfluidic device over two generations, with a cell viability rate in excess of 90%. Sister cells are successfully isolated after division and extracted within a 500 nL volume, which was demonstrated to be compatible with downstream single-cell RNA sequencing analysis.

## Introduction

One of the biggest challenges in quantitative biology is to better understand the decision-making process of cells. Over the past 20 years, a change in the scale of investigation from cell populations to the single-cell level has already brought numerous insights of such processes [1–3]. The primary benefit of performing experiments at the single-cell level is the ability to reveal the underlying transcriptional heterogeneity of both normal and pathological cells [4, 5]. Furthermore, single-cell studies have already provided evidence that gene expression variability is a property of cell fate decision making [3, 6].

available on the Git repository at https://gitbio.ens-lyon.fr/cfournea/sincity.

**Funding:** This work was supported by funding from the French agency ANR (SinCity; ANR-17-CE12-0031). The funders had no role in study design, data collection and analysis, decision to publish, or preparation of the manuscript.

**Competing interests:** The authors have declared that no competing interests exist.

Cellular differentiation is the process by which any pre-committed cell acquires its identity, and can be viewed as a dynamic process wired by the underlying gene regulatory network (GRN). Cells can be thought of as "moving particles" within a landscape, with the cell state space shaped by the GRN state [7]. According to this view, within this landscape, points of stability are referred as "steady states" and can be represented by attraction wells. Cells can escape their self-renewing steady state through a rise in gene expression variability and then explore freely, to some extent, the landscape to finally reach a new state of equilibrium; the differentiated state [7]. Single-cell analysis of *in vitro* and *in vivo* differentiation models have confirmed that this cellular process is indeed characterized by a global rise in gene expression variability [8–12]. That said, the way that gene expression variability is established across cell generations is still poorly understood. Such a fundamental question is likely to be of critical importance as it seems to be a conserved phenomenon across both biological systems and species [13–16]. Indeed, at the organism scale, during differentiation, cells must maintain their lineage identity through mitosis and eventually reach their differentiation state. Based on recent studies, support for this state memory comes from the inheritance of mRNA levels from mother cells to daughter cells [13]. This transmission is, with high probability, supported by the inheritance of epigenetic modifications allowing the maintenance of gene-specific transcription levels over cell divisions [16, 17]. Recently, it has been noted that in some genes, in which expression is variable amongst an isogenic cell population, expression is correlated between genealogically related cells [13, 14]. For some of these "memory genes", the correlation in expression may last for tens of generations. These data, gathered on self-renewing cells, imply a gene-specific transcriptional memory over several cell generations [13].

We recently developed experimental methods to recover related cells after one (first generation) and two (second generation) cell divisions, in order to investigate how cells reconcile the constraints of transcriptional memory and the rise in gene expression variability during the differentiation process [18]. Transcriptomics comparisons of self-renewing and differentiating sister and cousin cells indicated that transcriptional memory is gradually erased as differentiation proceeds. While (non-genetic) fluorescent barcoding techniques allow for the identification and tracking of individual cells and their lineage information for up to two cell divisions, it becomes challenging to extend this analysis to subsequent generations due to the difficulty in achieving high levels of fluorescent multiplexing [18]. Whilst other approaches do allow cell-tracking over multiple cell generations coupled with transcriptomics analysis, they require heavy genetic modifications (not compatible with the life span of primary cells) and do not provide the capability to track cell proliferation at the resolution of a single-cell division [19–21]. In contrast, microfluidic tools are recognized as being adept at performing single-cell manipulations [22], including the study of gene expression at the single-cell level [23, 24]. Moreover, microfluidic systems are well-suited to controlling heat and mass transfer, in a rapid and precise manner, and since they can be easily integrated with optical detection systems and imaging modalities, long-term tracking of cellular behavior becomes simple [25]. At a fundamental level, microfluidic cell culture systems have many advantages over conventional cell culture methods, including low reagent consumption, multiplexed operation and easy automation of cell culturing tasks [26]. Accordingly, the ability to monitor single-cell lineages and analyze differences between sister cells post division becomes possible, without needing to genetically modify mother cells.

Recently, several microfluidic-based cell culture systems for tracking cell lineage have been reported. For example, Kimmerling *et al.* used parallelized trapping structures to track the lineage of murine CD8$^+$ T-cells and lymphocytic leukemia cell lines [14]. Specifically, cells trapped in individual hydrodynamic traps are grown in a serpentine-shaped parallel microchannel network. After division, sister cells are separated using fluid flow through traps and

extracted via the device outlet. Although the device could be used to track cell lineage over multiple generations, fluid flow conditions and hydrodynamic trap geometries must be optimized for each cellular population. Additionally, it is not possible to address divided cells in an independent manner, and thus extracting specific sister cells is challenging. Other microfluidic approaches have been used to track and extract targeted cells from culture [27], but these almost always require extraction volumes (a few microliters) that are far too large for downstream transcriptomics analysis. Conversely, other approaches, such as those based on Fluidigm's Polaris system[28], allow single-cell transcriptomics measurements, but cannot track cell lineage over multiple generations. Accordingly, there remains a pressing and unmet need for an automated experimental platform that can perform both cell lineage tracking and single-cell extraction within volumes less than 1μL. To this end, we now describe the design, fabrication and development of an automated image-based microfluidic platform for tracking non-adherent single-cell lineage. Essential characteristics of the system include: (i) integrated microfluidic chambers for single-cell trapping, (ii) the ability to monitor cell growth over extended time periods, (iii) the ability to separate sister cells after division, (iv) facile reallocation of sister cells to monitor second and third division events and (v) extraction of cells for downstream transcriptomics analysis, using MARS-seq [29], a UMIs (Unique Molecular Identifier) and plate-based single-cell RNAseq protocol.

## Materials and methods

### Microscope incubator setup

To monitor cell proliferation *in vitro*, *in vivo* environmental conditions must be mimicked using a microscope placed inside an incubator. An inverted microscope (Eclipse Ti-E, Nikon, Egg, Switzerland) was enclosed within a custom-designed polycarbonate incubation box (Life Imaging Services, Basel, Switzerland) to provide optimum (5% $CO_2$ and 95% humidity, at 37°C) proliferation conditions. The box was then connected to an air-heater (Life Imaging Services, Basel, Switzerland). An in-house $CO_2$ chamber connected to a 5% $CO_2$ mixture tank (PanGas, Dagmersellen, Switzerland) with electronic flow control (Red-y, **Vögtlin Instruments GmbH**, Muttenz, Switzerland) was attached to the microfluidic device on a motorized *xy* translation stage (Mad City Labs GmbH, Kloten, Switzerland). An optical shutter was controlled by the ProScan III automation system (ProScan III, Prior Scientific Instruments GmbH, Jena, Germany) and used to regulate light exposure. A scientific complementary metal-oxide-semiconductor (sCMOS) camera (pco edge, PCO GmbH, Kelheim, Germany) in conjunction with a Plan Fluor 10X/0.3 NA objective (Nikon, Egg, Switzerland) was used to image cells for periods between 24 and 48 hours. A flow EZ™ pressure-based flow controller (Fluigent Deutschland GmbH, Jena, Germany) was used to deliver cells and reagents into the microfluidic device. MH1 solenoid valves (Festo AG, Lupfig, Switzerland) were incorporated within the microfluidic device and used to manipulate cells and automate the whole experimental process via a custom-developed MATLAB® code.

### Microfluidic device design and operation

The two-layer microfluidic device was designed to trap and allow proliferation of cells in a controlled manner. Fluid flow within the microfluidic device was generated using polydimethylsiloxane (PDMS)-based pneumatic microvalves [30]. The microfluidic device consists of a control layer and a fluidic layer, each consisting of a network of channels. The control layer is located above or below the fluidic layer and can be deformed so as to establish or terminate flow. Such valves can be designed to be "push-up" or "push-down" in nature, depending on the relative locations of the control and fluidic layer. Push-up valves are more desirable for

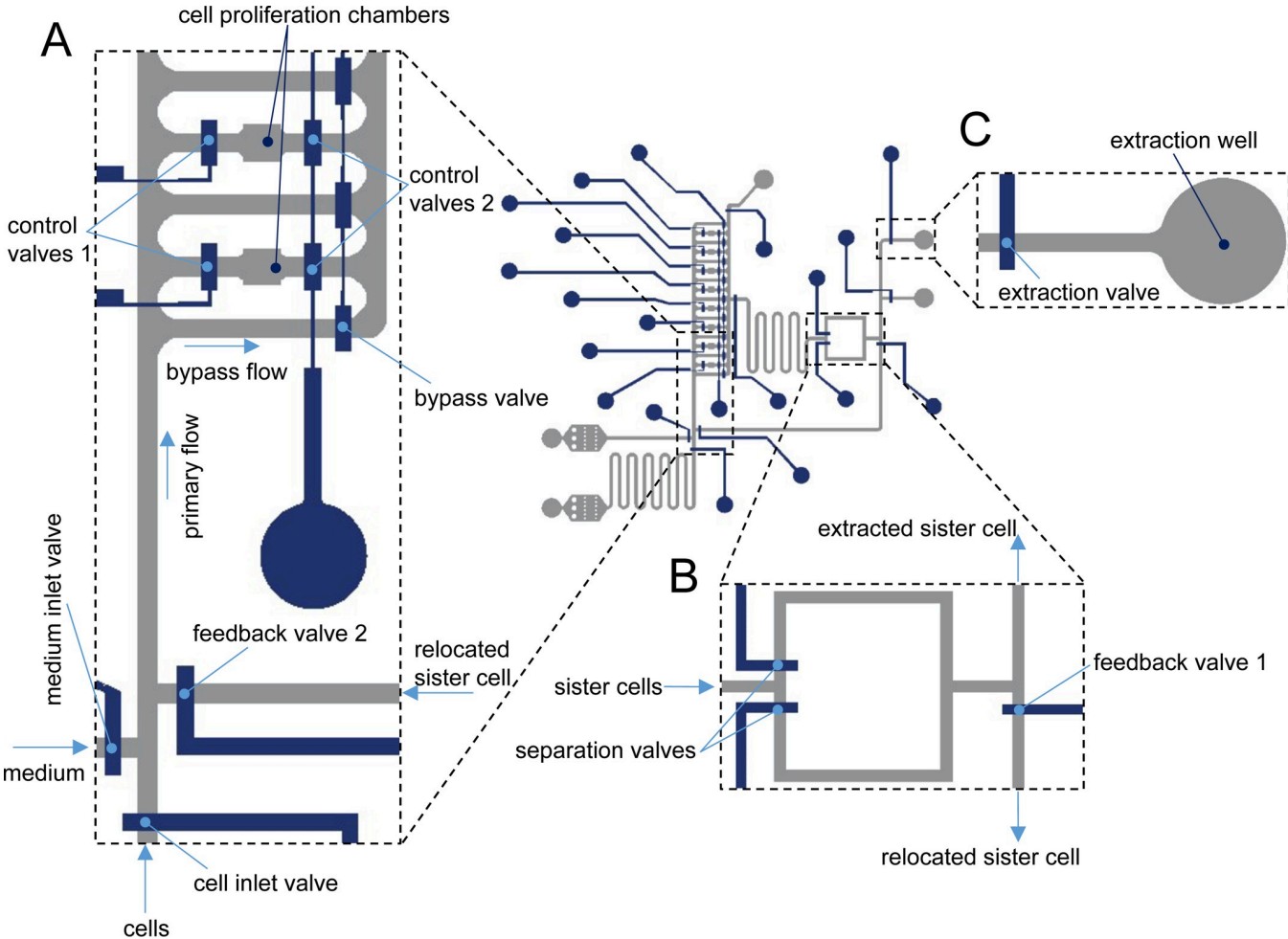

**Fig 1. Microfluidic single-cell processing platform and experimental workflow.** (A) The microfluidic device consists of cell inlet for delivering cells into the chambers, medium inlet for supplying fresh medium into the chambers after trapping single-cells, 8 individually addressable proliferation chambers, (B) a valve-based junction for the separation of the sister cells after division with a feedback channel that allows relocation of the sister cells after division from the separation area to the cell trapping chambers and (C) extraction wells for the collection of the sister cells. The workflow of the device comprises trapping of a single-cell inside a growth chamber, cell growth and division, separation of the sister cells after division and extraction of the individual sister cells for downstream transcriptome analysis.

applications involving eukaryotic cell manipulations within deeper fluidic channels, since they offer lower leakage flow compared with push-down valves. Push-down valves are more suitable when different materials are needed as a substrate material for microfluidic device instead of PDMS, for example when molecules are patterned on a glass slide [31]. In the current device, we used a push-up valve structure, since the device was exclusively intended for culturing and manipulating eukaryotic cells.

The two-layer microfluidic device integrates eight chambers for the long-term monitoring (> 24 hours) and tracking of sister stem cells over two generations (**Fig 1**). Single-cells were trapped inside proliferation chambers using control valve 1, which upon actuation, prevents fluid from entering the trapping region. Delivery of fresh cell medium to trapped cells is accomplished by opening bypass flow channels on each side of each chamber and control valve 2. The medium delivery process starts with a primary medium flow, firstly divided into many bypass flow paths. As noted, fluid flow through the bypass channels is regulated using

control valves, which maintain a constant circulation of fresh medium around the cell trapping chambers when open (**Fig 1A**). Sister cells separation, relocation of cells to new trapping chambers and single cell extraction were also performed within the device and are shown in **Fig 1B and 1C**. Specifically, after division, sister cells are manoeuvred into a separation zone that incorporates two control valves. Actuation of one of these valves ensures that one of the cells can be driven towards the extraction area, while the other cell will remain trapped; therefore, sister cells can be separated (**Fig 1B**). The feedback channel allows relocation of sister cells after division from the separation zone into the cell trapping chambers. Sister cells separated after division flow through the feedback channel upon actuation of the control valves (**Fig 1B**) and are subsequently placed in individual trapping chambers. After subsequent division events, new sister cells can be separated and either extracted from the device or relocated back to a trapping chamber for analysis of the third generation. The extraction area includes two independently addressable, 1 mm diameter and 3 mm depth open wells for the collection of sister cells (**Fig 1C**). The current microfluidic device integrates eight chambers, and thus allows monitoring of up to three generations from a single-cell (from one parent cell to eight daughter cells).

## Cell culture

6C2 chicken erythroblast cells, transformed by the avian erythroblastosis virus (AEV) carrying a stably integrated mCherry transgene, were maintained in αMinimal Essential Medium (Thermo Fischer Scientific, Basel, Switzerland) complemented with 10% Fetal Bovine Serum (FBS, Life Technologies, Zug, Switzerland), 1% Normal Chicken Serum (Thermo Fischer Scientific, Basel, Switzerland) [32], 1% penicillin and streptomycin (10,000 U/ml, Thermo Fischer Scientific, Basel, Switzerland), 100 nM β-mercaptoethanol (Sigma-Aldrich, Buchs, Switzerland), and kept at 37˚C with 5% $CO_2$ in an incubator (New Brunswick Galaxy 170 S, Eppendorf, Schönenbuch, Switzerland).

T2EC cells were extracted from the bone marrow of white leghorn chicken embryos (INRA, Tours, France) [33]. The cells were cultured in αMinimal Essential Medium (Gibco), supplemented with 1 mM HEPES (Sigma-Aldrich), 10% Fetal Bovine Serum (FBS, BioWest), 1% Penicillin-Streptomycin (10,000 U/mL, Gibco), 100 nM β-mercaptoethanol (Sigma-Aldrich), 1 mM dexamethasone (Sigma-Aldrich), 5 ng/mL transforming growth factor-alpha (TGF-α, Peprotech) and 1 ng/mL transforming growth factor-beta (TGF-β, Peprotech), and kept at 37˚C with 5% $CO_2$ in an incubator.

## ScRNA-seq library preparation

Single-cell RNA library preparation was performed using an adapted version of the MARS-seq protocol (Massively parallel single-cell RNA sequencing) [29], as described in detail elsewhere [34]. The complete library consisted of 10 microfluidics-sorted cells and 86 FACS-sorted cells.

## RNA sequencing

Sequencing was performed on a Nextseq500 sequencer (Illumina, IGFL sequencing platform (PSI), Lyon, France), with a custom paired-end protocol to avoid a decrease in sequencing quality on read1 due to a high number of T bases added during polyA reading (130pb on read1 and 20pb on read2), and a targeted depth of 200 000 raw reads per cell.

## Data pre-processing

Fastq files were pre-processed using an in-house bio-informatics pipeline on the Nextflow platform (Seqera Labs, Barcleona, Spain) [35], as described elsewhere [34]. Briefly, the first

step removed Illumina adaptors sequences. The second step de-multiplexed the sequences according to their plate barcodes. Next, all reads containing at least 4 T bases following the cell barcode and UMI sequences were kept. Using the UMItools whitelist, the cell barcodes and UMI were extracted from the reads. The sequences were then mapped on the reference transcriptome (Gallus GallusGRCG6A.95 from Ensembl) and UMIs were counted. Finally, a count matrix was generated.

## Quality control and data filtering

All analyses were carried out using R software (version 4.2.1 [36]). Cells were filtered based on several criteria: read number, gene number, count number and ERCC content. For each criterion the cut off values were determined based on the SCONE [37] pipeline and calculated as follows:

Mean (criterion value) - 3*sd (criterion value). After the cell filtering step there remained 7 chip-sorted cells and 82 FACS sorted control cells. Among the chip-sorted cells, 4 of these were sister cells (two couples of cells arising from the mitosis of the same mother cells), and 3 were orphan cells, meaning cells for which the other sister cell was eliminated from the dataset due to poor quality, either from lack of recovery or insufficient lysis. Based on work by Breda *et al.* [38], genes were kept in the data set if they were expressed on average in every cell (in average 1 UMI per cell).

## Normalization

The filtered matrix was normalized using SCTransform from the Seurat package [39] and corrected for sequencing depth.

## UMAP

Dimensionality reduction and visualization was performed using UMAP [40] default parameters.

## Results

### Mother cell capture and first division

The microfluidic device was used to process two different (non-adherent) cell models, 6C2 and T2EC. 6C2 cells are transformed erythrocytic progenitors and constitutively express a mCherry transgene. T2EC cells are primary erythrocytic progenitors, extracted from chicken bone marrow (see Materials and Methods). Experiments on both cell models were performed independently. Cells were introduced using the pressure-based flow controller at a concentration of $10^6$ cells/mL suspension, with single-cells being trapped individually in trapping chambers (i.e. one cell per chamber), as shown in **Fig 2**. Trapped (unrelated) single-cells, referred to as mother cells, were then monitored over a period of 24 hours. Cell divisions were observed for both single 6C2 cells and T2EC cells (**Fig 2**) after approximately 6 and 10 hours of culture within the microfluidic device. The average cell division rate for both models (over a sample of 20 single-cells) matched the expected division rate of bulk 6C2 and T2EC, and in a time frame known for division of those cells in regular culture conditions [32].

### Sister cells separation

After cell division occurred, sister cells in a given chamber were separated. In this regard, it is noted that 6C2 sister cells spontaneously separated after mitosis, while T2EC cells stayed attached to each other, thus necessitating enzymatic dissociation. Specifically, we temporarily

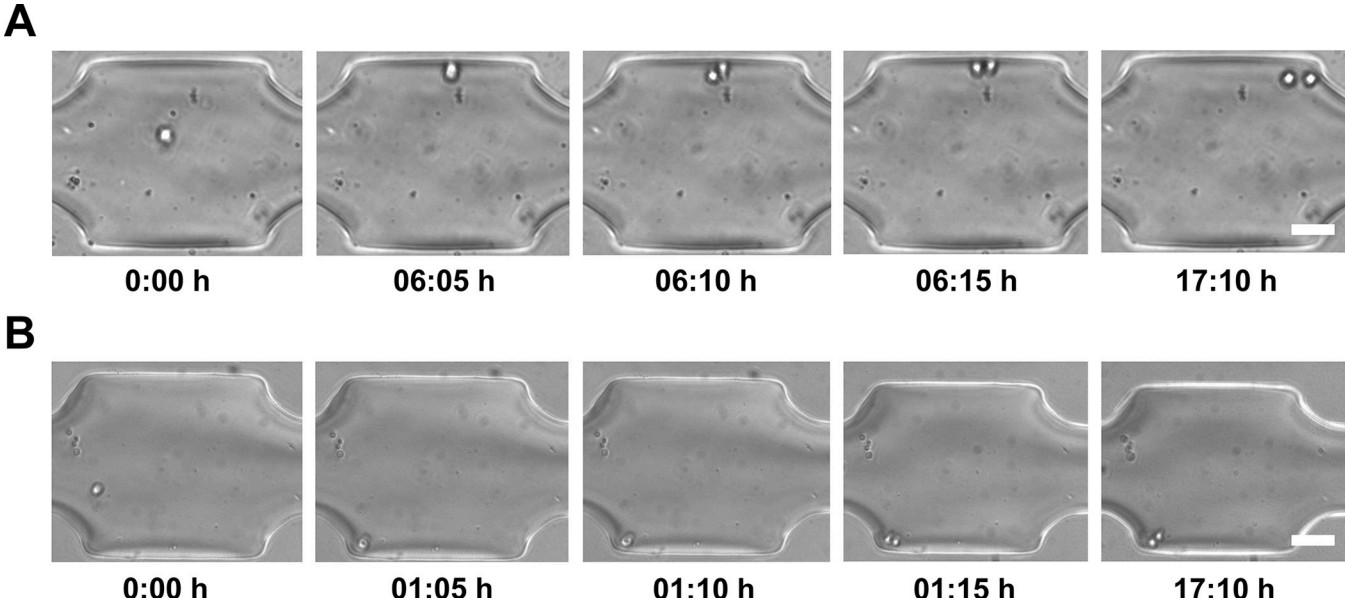

**Fig 2. Single-cell proliferation experiments.** (A) Single 6C2 cells, (B) Single T2EC cells were trapped and monitored over a period of 24 hours. Time lapse brightfield images for one chamber were acquired in every 5 minutes. The brightfield images show that cell full division occurs in each chamber within 20 minutes. The scale bar is 50 μm.

replaced the culture medium with Accutase®, an enzymatic complex of marine origin, presenting proteolytic and collagenolytic activity and less toxic than Trypsin, thus ensuring cell dissociation under mild conditions. Accordingly, T2EC cell pairs were dissociated by flowing Accutase® (ready to use - 1X) through the chamber at 37˚C for a period of 30–45 minutes, with separation being monitored by direct brightfield observation (**Fig 3**). Accutase® self-inactivates after 30–45 minutes at 37˚C, and therefore there is no need flush the solution out after dissociation has occurred.

## Sister cells relocation and second division

We next separated the sister cells after the first cell division, and relocated each sister in a different chamber, in order to allow secondary cell division events. Using the T2EC cell model, we located a chamber where a division had occurred (i.e. observation of a cell doublet). The sister cells, resulting from the first division of the mother cell, were separated using Accutase as described above and were moved towards the separation zone which incorporates two control valves (**Fig 1B**). The fluid flow was precisely controlled with the pressure pump by applying a pressure of approximately 10 mbars. Accordingly, sister cells moved into a long serpentine channel that connects cell chambers to the separation area in a controlled manner. When the cells were flowing through the serpentine channel to the separation area a sufficient cell-to-cell distance or spacing (a few hundreds of microns) always occurred. In the separation area, actuation of one of the control valves ensured that one of the cells was driven towards relocation area (or extraction area), while the other cell remained trapped (**S1 Video**). The feedback channel allowed relocation of sister cells after division from the separation zone into the cell trapping chambers. Sister cells separated after division circulated through the feedback channel upon actuation of the control valves and fine control of flow pressure, and were subsequently placed in individual trapping chambers (**Fig 3**).

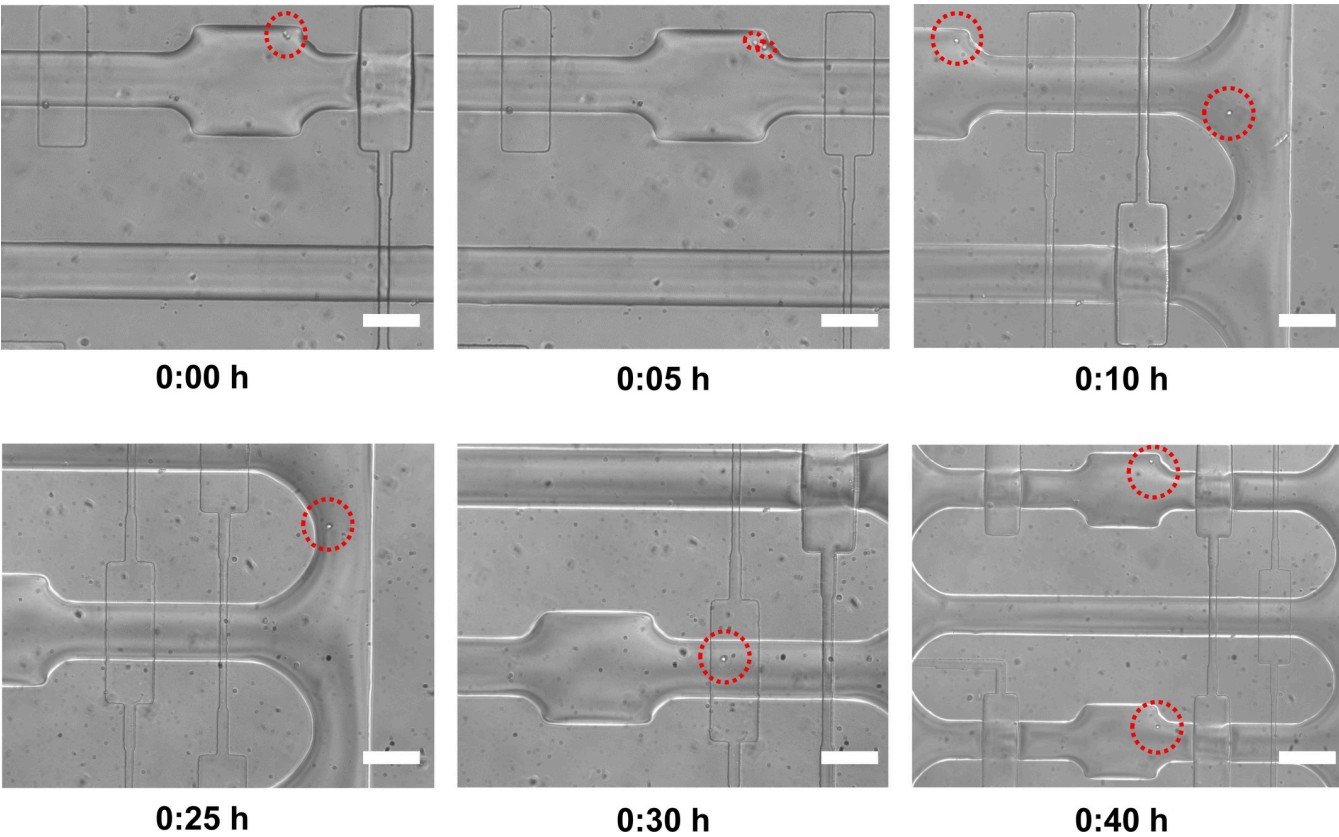

**Fig 3. Single T2EC cell proliferation and sister cell relocation.** A single T2EC cell was trapped and monitored by time-lapse brightfield microscopy, with images being acquired every 5 minutes. During sister cell relocation, the first sister is kept in the initial chamber, with the second sister being moved in a new chamber, by applying 10 mbar of pressure from the medium inlet which allows precise control of the single-cell movement. The scale bar is 50 µm.

## Sister cells extraction

Extraction experiments were only performed on 6C2 cells, since they consist of transformed erythrocytic progenitors and constitutively express a mCherry transgene. This allows facile monitoring of the cell extraction process via fluorescence imaging. As noted, the most challenging task within the experimental workflow is the extraction and collection of selected cells within a fluid volume no larger than 500 nL. Such a requirement is set by the need to ensure compatibility with downstream scRNA-seq analysis [41]. Indeed, the first step in scRNA-seq library construction involves reverse transcription of all mRNAs from each individual cell. This process must be carried out in a very small reaction volume (<4 uL) since it is prone to molecular inhibition due to the high number of proteins present in the culture medium used for cell isolation. The volume in which the cell should be isolated must be kept as low as possible (below 20% of the total reaction volume) and be reproducible for each isolated cell, to minimize the variability in efficacy of the reverse transcription from one cell to another.

Each selected cell was delivered to the pre-punched extraction well by applying 10 mbar of pressure from the medium inlet, resulting in a cellular velocity of 10 µm/s, with fluorescence imaging being used to track single-cells after their delivery into the extraction well. Next, single-cells were extracted from the device using a thin graduated capillary tube (**Fig 4A**). The glass capillary tube was inserted into the well to extract the cell via capillarity. The extraction area includes two independently addressable, 1 mm diameter and 3 mm depth open wells for

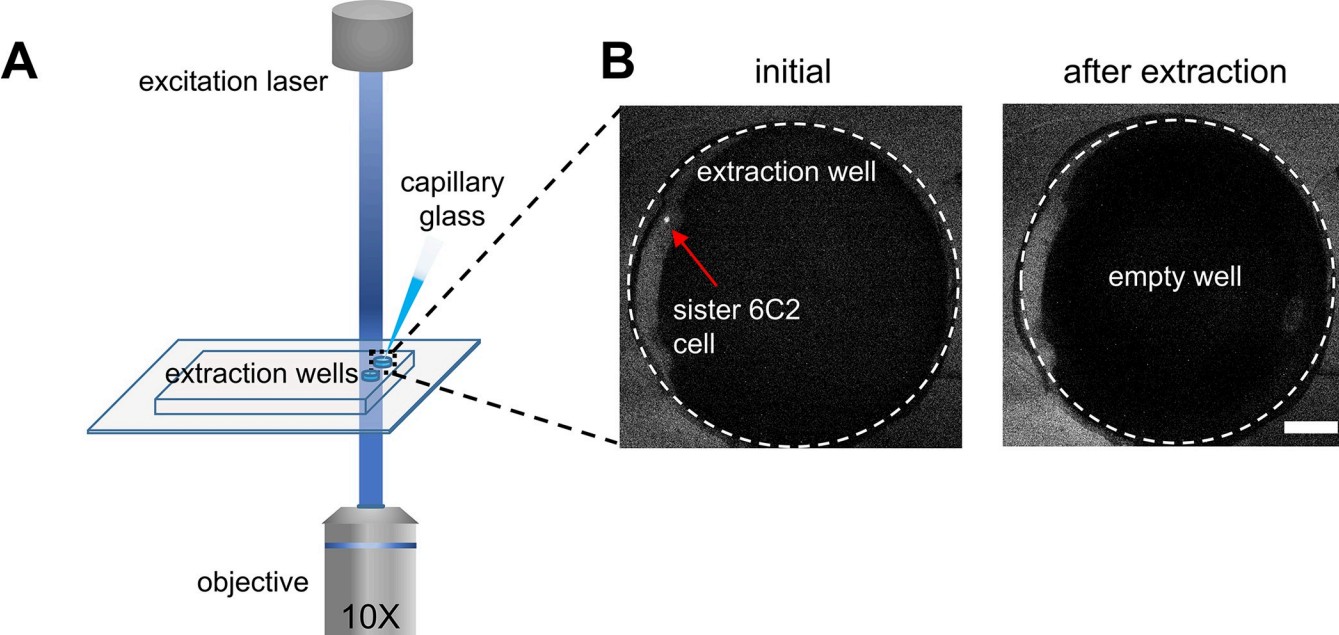

**Fig 4. Sisters cell extraction.** (A) A single 6C2 sister cell is monitored using fluorescence imaging in the extraction well. Manual extraction of this cell is performed using a small glass capillary. (B) Fluorescence images of the extraction well before and after the extraction of a single sister cell. The scale bars are 100 μm.

the collection of sister cells (Fig 1C). Cells were led to the pre-punched extraction wells, then, the capillary glass was plugged in the well. Since the tip diameter of the capillary glass is smaller than the well diameter, the capillary glass can be easily introduced into the well. As aforementioned, the driven pressure is very low (~10 mbars) which results in a cell velocity of 10 μm/s allowing an easy tracking of the cell inside the extraction well. After a single cell was driven inside the well, fluid flow was stopped during the extraction process and the single cell was collected via the capillary flow. Therefore, the fluid, included the single cell, did not flow out of the well. Additionally, the well has a reservoir volume of 2.5 μL, significantly larger than the extraction volume and thus the fluid containing a single cell would not flow out during the extraction process.

Furthermore, the extraction volume could be precisely controlled inspecting the graduations on the capillary, and fluorescence imaging ensured that a desired cell had been successfully extracted (**Fig 4B**). Significantly, this method proved to work successfully for extraction volumes less than 500 nL, and was therefore compatible with downstream scRNA-seq analysis.

### Capture of 6C2 mother cells, first division and extraction of sister cells for scRNA-seq downstream analysis

We performed a proof-of-concept experiment on 6C2 cells, in which five mother cells were isolated in independent chambers. Each chamber was then monitored over an extended period of time, allowing observation of first division events by time-lapse brightfield microscopy. After division, the resulting sister cells were extracted from the microfluidic device, as described previously. After extraction, each of the ten isolated sister cells was directly transferred in lysis buffer.

Before constructing the library, 86 6C2 FACS-sorted single-cells, from a population where relationships between the cells were unknown, were barcoded and added to the cell pool

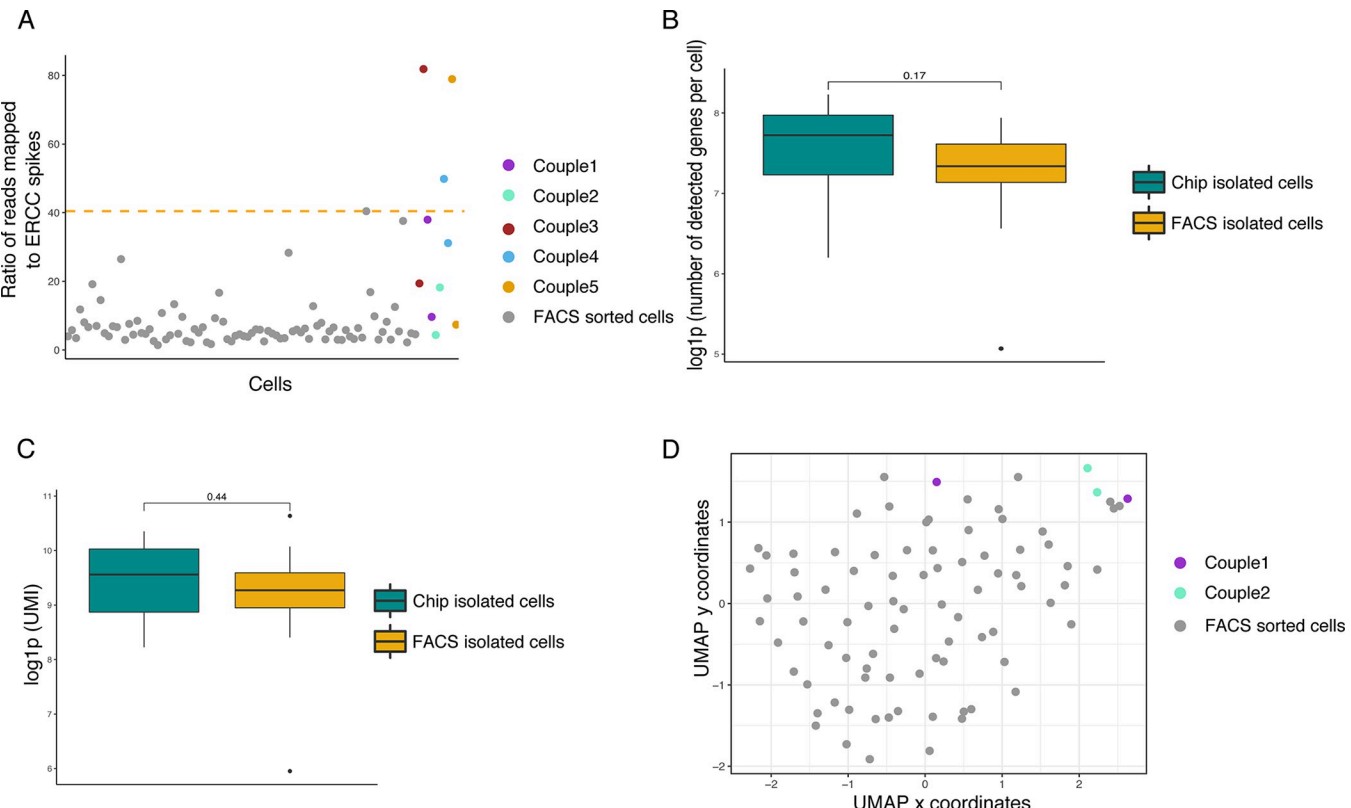

**Fig 5. scRNA-seq data vizualisation.** (A) Plot of the ratio of ERCC mapped in each cell. The orange line represents the cut off value; cells positioned higher than the cut off are discarded. (B) Boxplot showing the number of detected genes per cell, sorted with conventional FACS or cultured and isolated using the microfluidic platform. A Wilcoxon rank-test was performed. (C) Boxplot of log(UMIs) number per cell sorted with conventional FACS or cultured and isolated using the microfluidic platform. A Wilcoxon rank-test was performed. (D) UMAP dimensions reduction and projection of the cells. Only complete couples of sister cells were kept for the analysis. Chip-cultured cells are coloured and grouped by lineage and FACS sorted cells are grey.

experiment. FACS-sorted cells were used as controls, since FACS sorting is the reference method for isolating single-cells for subsequent scRNA-seq analysis. Following cell isolation, within each sample, ERCC spikes (External RNA Controls Consortium [42]) were added. ERCCs consist of 92 different synthetic RNAs species and are used as experimental controls, since they are inserted in a known concentration and will undergo all the steps of library construction, as do the cellular mRNAs. Each cell's mRNA and associated ERCC were barcoded with a unique cell barcode and UMI, by reverse transcription using RT primers for which the cell barcode sequence was known (see Methods section).

The scRNA-seq library, consisting of the 10 microfluidically-sorted single-cells and the 86 FACS-sorted single-cells, was then generated using a protocol detailed elsewhere [34] and sequenced as described previously. As noted, raw sequencing data were processed on using an in-house bio-informatics pipeline, filtered and normalized. As a quality control step, the ratio of ERCC counts over cellular mRNA counts was compared between FACS-isolated cells and microfluidically-isolated cells (**Fig 5A**). If this ratio is high, cellular mRNAs are in low number, indicating that either the cell was not captured properly, lysis was incomplete, or the cell was stressed at the time of isolation (and thus its mRNAs were starting to degrade).

After data quality filtering, among the 10 microfluidically-sorted cells, a total of 7 passed quality filters; the three "poor quality" cells were most likely damaged or not recovered, as shown by their high content of ERCC spikes RNA compared to the content of cellular mRNA

(Fig 5A). Among the remaining seven cells, two complete sister cell couples were recovered. The sister cells isolated using our microfluidic platform displayed the same amount of mean detected genes per cell and mean UMIs, which reflects the total number of molecules per cells, as the control FACS-sorted cells (**Fig 5B and 5C, respectively**). The application of UMAP dimensionality reduction and projection revealed that chip-cultured and isolated sister cells did not significantly differ from control FACS-sorted cells, as shown by the fairly uniform repartition of all cells within the graph (**Fig 5D**).

## Conclusions and discussion

In this study, we have described the development of a multilayer microfluidic device and experimental workflow for tracking non-adherent cell divisions at the single-cell level. The microfluidic platform is able to concurrently trap single-cells in eight independently controlled proliferation chambers, isolate sister cells after division and extract them for downstream analysis. We have demonstrated that the system is capable of tracking cells over at least two generations using two different cell models (i.e. a cell line and primary cells). The complete platform incorporates semi-automated cell loading, long-term cell monitoring and cell extraction. Characterisation experiments confirmed that both 6C2 (chicken erythroleukemia cell line) and T2EC (chicken primary erythrocytic progenitors) cells proliferated inside the chip, with a viability rate higher than 90%. Divided cells were separated and placed inside the 500 nL-volume extraction chambers, which were compatible with downstream scRNA-seq analysis. Our general method allows the recovery of selected single-cells and the extraction of genealogical information of the cell, while providing the same data quality required for subsequent scRNA-seq analysis, as provided by regular FACS sorting. More generally, the developed system provides a robust and automated platform for single-cell lineage tracking studies at the single-cell resolution, and can be used to track non-adherent cells, including cell lines and primary cells. In the future, we expect that the device will be highly useful in performing perturbation experiments, including induction of differentiation and gene expression modulation using drugs, by changing culture reagents during the culture process. Moreover, analytical throughput can be significantly enhanced by increasing the number of parallel proliferation chambers per device, automation of single-cell trapping and automatic detection of cell division and relocation.

## Supporting information

**S1 Video. Video of sister cells exiting the chambers area and moving toward the separation area.**
(MP4)

## Acknowledgments

We thank the computational center of IN2P3 (Villeurbanne/France) and Pôle Scientifique de Modélisation Numérique (PSMN, Ecole Normale Supérieure de Lyon) where computations were performed. We acknowledge the contribution of the AniRA-Cytométrie core facility of SFR BioSciences (UAR3444/US8). We thank the BioSyL Federation and the LabEx Ecofect (ANR-11-LABX-0048) of the University of Lyon for inspiring scientific events.

## Author Contributions

**Conceptualization:** Mahmut Aslan Kamil, Camille Fourneaux, Andras Paldi, Sandrine Gonin-Giraud, Andrew J. deMello, Olivier Gandrillon.

**Funding acquisition:** Andras Paldi, Andrew J. deMello, Olivier Gandrillon.

**Investigation:** Mahmut Aslan Kamil, Camille Fourneaux, Alperen Yilmaz, Stavrakis Stavros, Romuald Parmentier.

**Methodology:** Mahmut Aslan Kamil, Camille Fourneaux.

**Supervision:** Andras Paldi, Sandrine Gonin-Giraud, Andrew J. deMello, Olivier Gandrillon.

**Writing – original draft:** Mahmut Aslan Kamil, Camille Fourneaux.

**Writing – review & editing:** Mahmut Aslan Kamil, Camille Fourneaux, Andras Paldi, Sandrine Gonin-Giraud, Andrew J. deMello, Olivier Gandrillon.

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
