## [Decision Letter · Decision Letter 0]

18 May 2023

PONE-D-23-09681An Image-Guided Microfluidic System for Single-Cell Lineage TrackingPLOS ONE

Dear Dr. Gandrillon,

Thank you for submitting your manuscript to PLOS ONE. After careful consideration, we feel that it has merit but does not fully meet PLOS ONE’s publication criteria as it currently stands. Therefore, we invite you to submit a revised version of the manuscript that addresses the points raised during the review process.

We look forward to receiving your revised manuscript.

Kind regards,

Hon Fai Chan, PhD

Academic Editor

PLOS ONE

Journal Requirements:

   "This work was supported by funding from the French agency ANR (SinCity; ANR-17-CE12-0031)."

    "NO"

Additional Editor Comments:

Changes required for acceptance:

Please address the reviewers' comments by providing detailed description of the processes of cell separation and cell extraction for sequencing. In addition, please correct the typos found in the manuscript. 

Reviewers' comments:

Reviewer's Responses to Questions

**Comments to the Author**

1. Is the manuscript technically sound, and do the data support the conclusions?

Reviewer #1: Yes

Reviewer #2: Yes

2. Has the statistical analysis been performed appropriately and rigorously? 

Reviewer #1: Yes

Reviewer #2: Yes

3. Have the authors made all data underlying the findings in their manuscript fully available?

Reviewer #1: Yes

Reviewer #2: Yes

4. Is the manuscript presented in an intelligible fashion and written in standard English?

Reviewer #1: Yes

Reviewer #2: Yes

5. Review Comments to the Author

Reviewer #1: In this paper, the authors have developed a multilayer microfluidic device and an experimental workflow for tracking non-adherent cell divisions at the single-cell level. The microfluidic platform could trap single-cells in eight independently controlled proliferation chambers, isolate sister cells after division and extract them for downstream analysis. They demonstrated that the platform could track cells over at least two generations. In comparison with related reports, the design is original and novel and the single-cell extraction volume is down to 500 nL. However, there are typos in the context need to be corrected like on page 3, line 42 a period is missing and so on.

Reviewer #2: This paper present a setup of microfluidic system for monitoring cell division and isolation of sister cells for post process such as single cell sequencing. The technique looks fine in general. Some revisions or clarifications are needed before publication for better reproducibility.

1. After division, the sister cells are close to each other even after dissociation. The author mentioned the sister cells were to separate into two different chambers. Please give detailed operation process on how to move one cell into another chamber while keep the other cell stay, as demonstrated in Fig. 3. Wouldn’t both of the cells move with the flow to new places?

2. Single cells were extracted with capillary glass for sequencing. How was the capillary glass inserted into the well? Through PDMS slab or through any hole punched in PDMS? I don’t find description on this part. Directly plugging a capillary glass through PDMS sounds quite difficult, especially into a micron sized well precisely. However, if a hole was pre-punched, wouldn’t the fluid flow out due to the driven pressure? Please describe in details for clarification.

6. PLOS authors have the option to publish the peer review history of their article (what does this mean?). If published, this will include your full peer review and any attached files.

Reviewer #1: No

Reviewer #2: No

---

## [Author Response · Author response to Decision Letter 0]

27 Jun 2023

Journal Requirements:

Done

 "This work was supported by funding from the French agency ANR (SinCity; ANR-17-CE12-0031)."

Done

 "NO"

Done

Done

5. Please include your full ethics statement in the 'Methods' section of your manuscript file. In your statement, please include the full name of the IRB or ethics committee who approved or waived your study, as well as whether or not you obtained informed written or verbal consent. If consent was waived for your study, please include this information in your statement as well. 

Not applicable (no human subjects involved).

Review Comments to the Author

Reviewer #1: In this paper, the authors have developed a multilayer microfluidic device and an experimental workflow for tracking non-adherent cell divisions at the single-cell level. The microfluidic platform could trap single-cells in eight independently controlled proliferation chambers, isolate sister cells after division and extract them for downstream analysis. They demonstrated that the platform could track cells over at least two generations. In comparison with related reports, the design is original and novel and the single-cell extraction volume is down to 500 nL. However, there are typos in the context need to be corrected like on page 3, line 42 a period is missing and so on.

This typo has been corrected. 

Reviewer #2: This paper present a setup of microfluidic system for monitoring cell division and isolation of sister cells for post process such as single cell sequencing. The technique looks fine in general. Some revisions or clarifications are needed before publication for better reproducibility.

1. After division, the sister cells are close to each other even after dissociation. The author mentioned the sister cells were to separate into two different chambers. Please give detailed operation process on how to move one cell into another chamber while keep the other cell stay, as demonstrated in Fig. 3. Wouldn't both of the cells move with the flow to new places?

To answer this comment the following modifications were made in the manuscript:

The following statement was added line 261-274:

The sister cells, resulting from the first division of the mother cell, were separated using Accutase as described above and were moved towards the separation zone which incorporates two control valves (Fig 1B). The fluid flow was precisely controlled with the pressure pump by applying a pressure of approximately 10 mbars. Accordingly, sister cells moved into a long serpentine channel that connects cell chambers to the separation area in a controlled manner. When the cells were flowing through the serpentine channel to the separation area a sufficient cell-to-cell distance or spacing (a few hundreds of microns) always occurred. In the separation area, actuation of one of the control valves ensured that one of the cells was driven towards relocation area (or extraction area), while the other cell remained trapped (S1 video). The feedback channel allowed relocation of sister cells after division from the separation zone into the cell trapping chambers. Sister cells separated after division circulated through the feedback channel upon actuation of the control valves and fine control of flow pressure, and were subsequently placed in individual trapping chambers (Fig 3).

We also added supplementary video 1 showing sister separation in the separation area for relocation or extraction. The following statement was added line 570:

S1 : Video of sister cells exiting the chambers area and moving toward the separation area.

2. Single cells were extracted with capillary glass for sequencing. How was the capillary glass inserted into the well? Through PDMS slab or through any hole punched in PDMS? I don't find description on this part. Directly plugging a capillary glass through PDMS sounds quite difficult, especially into a micron sized well precisely. However, if a hole was pre-punched, wouldn't the fluid flow out due to the driven pressure? Please describe in details for clarification.

To answer this comment the following modifications were made in the manuscript:

The selection paragraph was rewritten as follows (line 290-306):

Each selected cell was delivered to the pre-punched extraction well by applying 10 mbar of pressure from the medium inlet, resulting in a cellular velocity of 10 µm/s, with fluorescence imaging being used to track single-cells after their delivery into the extraction well. Next, single-cells were extracted from the device using a thin graduated capillary tube (Fig 4A). The glass capillary tube was inserted into the well to extract the cell via capillarity. The extraction area includes two independently addressable, 1 mm diameter and 3 mm depth open wells for the collection of sister cells (Fig 1C). Cells are led to the pre-punched extraction wells, then, the capillary glass is plugged in the well. Since the tip diameter of the capillary glass is smaller than the well diameter, the capillary glass can be easily introduced into the well. As aforementioned, the driven pressure is very low (~10 mbars) which results in a cell velocity of 10 µm/s allowing an easy tracking of the cell inside the extraction well. After a single cell was driven inside the well, fluid flow was stopped during the extraction process and the single cell was collected via the capillary flow. Therefore, the fluid, included the single cell, did not flow out of the well. Additionally, the well has a reservoir volume of 2.5 µL significantly larger than the extraction volume and thus the fluid containing a single cell would not flow out during the extraction process.

---

## [Decision Letter · Decision Letter 1]

3 Jul 2023

An Image-Guided Microfluidic System for Single-Cell Lineage Tracking

PONE-D-23-09681R1

Dear Dr. Gandrillon,

We’re pleased to inform you that your manuscript has been judged scientifically suitable for publication and will be formally accepted for publication once it meets all outstanding technical requirements.

Kind regards,

Hon Fai Chan, PhD

Academic Editor

PLOS ONE

Additional Editor Comments (optional):

The manuscript is now acceptable for publication.

Reviewers' comments:

Reviewer's Responses to Questions

**Comments to the Author**

1. If the authors have adequately addressed your comments raised in a previous round of review and you feel that this manuscript is now acceptable for publication, you may indicate that here to bypass the “Comments to the Author” section, enter your conflict of interest statement in the “Confidential to Editor” section, and submit your "Accept" recommendation.

Reviewer #1: All comments have been addressed

Reviewer #2: All comments have been addressed

2. Is the manuscript technically sound, and do the data support the conclusions?

Reviewer #1: Yes

Reviewer #2: Yes

3. Has the statistical analysis been performed appropriately and rigorously? 

Reviewer #1: Yes

Reviewer #2: Yes

4. Have the authors made all data underlying the findings in their manuscript fully available?

Reviewer #1: Yes

Reviewer #2: Yes

5. Is the manuscript presented in an intelligible fashion and written in standard English?

Reviewer #1: Yes

Reviewer #2: Yes

6. Review Comments to the Author

Reviewer #1: (No Response)

Reviewer #2: In the revised version, the authors have addressed all my concerns. I think it is good for publication.

7. PLOS authors have the option to publish the peer review history of their article (what does this mean?). If published, this will include your full peer review and any attached files.

Reviewer #1: No

Reviewer #2: No

---

## [Editor Report · Acceptance letter]

21 Jul 2023

PONE-D-23-09681R1 

An Image-Guided Microfluidic System for Single-Cell Lineage Tracking 

Dear Dr. Gandrillon:

I'm pleased to inform you that your manuscript has been deemed suitable for publication in PLOS ONE. Congratulations! Your manuscript is now with our production department. 

Kind regards, 

on behalf of

Professor Hon Fai Chan 

Academic Editor

PLOS ONE